# Trends of Nursing Research on Accidental Falls: A Topic Modeling Analysis

**DOI:** 10.3390/ijerph18083963

**Published:** 2021-04-09

**Authors:** Yeji Seo, Kyunghee Kim, Ji-Su Kim

**Affiliations:** Department of Nursing, Chung-Ang University, 84 Heukseok-Ro Dongjak-Gu, Seoul 06974, Korea; yejihj23@cau.ac.kr (Y.S.); kyung@cau.ac.kr (K.K.)

**Keywords:** accidental falls, topic modeling, nursing research, trend

## Abstract

This descriptive study analyzed 1849 international and 212 Korean studies to explore the main topics of nursing research on accidental falls. We extracted only nouns from each abstract, and four topics were identified through topic modeling, which were divided into aspects of fall prevention and its consequences. “Fall prevention program and scale” is popular among studies on the validity of fall risk assessment tools and the development of exercise and education programs. “Nursing strategy for fall prevention” is common in studies on nurse education programs and practice guidelines to improve the quality of patient safety care. “Hospitalization by fall injury” is used in studies about delayed discharge, increased costs, and deaths of subjects with fall risk factors hospitalized at medical institutions due to fall-related injuries. “Long-term care facility falls” is popular in studies about interventions to prevent fall injuries that occur in conjunction with dementia in long-term care facilities. It is necessary to establish a system and policy for fall prevention in Korean medical institutions. This study confirms the trends in domestic and international fall-related research, suggesting the need for studies to address insufficient fall-related policies and systems and translational research to be applied in clinical trials.

## 1. Introduction

Falls refer to unintentional changes in posture toward the floor or a lower position [1]. Falls represent a crucial patient safety issue in healthcare facilities worldwide and are often used as an index to evaluate the quality of nursing [2,3]. The incidence of patient falls varies among countries and medical institutions. In the United States, 3.3–11.5 falls are observed per 1000 hospitalization days, while in South Korea 400–700 falls annually have been reported [4,5,6,7,8,9]. Falls often have negative consequences, including physical injuries such as bleeding and fractures, psychological damage such as anxiety, fear, and loss of independence, and even death [10,11,12]. Furthermore, fall-related injuries increase socioeconomic burdens because of increased hospital stays and medical expenses [13,14]. Moreover, in attempts to prevent falls due to these socioeconomic issues, fall-related nursing research has been extensively carried out to date. This study details the current status of fall-related nursing research and its trends over time.

Various measures have been implemented on an international level to prevent falls in care facilities. In 2005, the Joint Commission International (JCI) listed falls as an essential standard item of international patient safety targets [15,16]. In 2008, the Centers for Medicare and Medicaid Services designated falls as a preventable injury, which further increased caution and interest in fall prevention [17]. In South Korea, falls have been listed as a mandatory item in the certification guidelines of the Korea Institute for Healthcare & Accreditation to reduce fall-related injuries [18,19]. Additionally, in 2013 and 2016, respectively, comprehensive nursing service laws and patient safety laws were enforced to prevent and manage falls [20,21].

Various nursing-related studies have also been conducted in and outside South Korea to develop nursing intervention plans to prevent falls. In particular, systematic literature reviews and meta-analyses have been conducted on previously published fall-related nursing studies, as well as various other studies on the definition [22], incidence [23], and risk factors of falls [24,25,26], fall risk assessment tools [27,28], and the effects of fall prevention programs [29,30,31,32]. However, the focus of many systematic literature review studies has been limited to specific areas such as the subjects, variables, environments, and interventions of studies. Thus, many studies could not be included, which limited the derivation of integrated results [33]. Further, since falls are caused by a combination of intrinsic and extrinsic factors [1,34], an integrated body of knowledge—such as interrelationships of issues and changes and trends over time—needs to be utilized in clinical practice [6].

Nurses are the key professionals responsible for fall prevention education and management for patient safety in hospitals and communities [35,36]. Further, the JCI emphasizes the importance of nursing in fall prevention by providing periodic fall risk assessments by nurses and additional nursing interventions [14]. Considering the increasing expectations from nurses to prevent falls, it would be purposeful to suggest the direction of future nursing studies for the development of effective fall prevention nursing interventions. This can be accomplished by evaluating the trends and courses of fall-related nursing studies conducted thus far, both in South Korea and internationally.

Recently, futurological studies using topic modeling as a big data analysis method to track changes in issues over time and predict future issues have gained interest [37,38]. Existing studies have also attempted to analyze research trends, such as the current status of studies and trends over time [39,40]. Analyses of abstracts and keywords have also been performed to examine research trends [41]. Topic modeling provides integrated information by identifying potential topics in the texts and analyzing the relationships between topics and their distribution [42]; this helps to microscopically identify key themes and relationships and macroscopically understand the flow and context of the main topics and the trend of topics by period [43,44]. Therefore, topic modeling is a valuable methodology for forming a body of knowledge on nursing studies that can be used to assess the pattern of texts and contextual structures through quantitative and qualitative analysis [45,46]. A fall is a sudden complex event that occurs by combining intrinsic and extrinsic factors and affects many aspects of individuals, society, and local communities [1,34]. It will be effective to analyze the flow and trends of big data of fall-related nursing research over time using topic modeling. Considering that studies related to social interest in falls will continue to increase in the future [47,48,49], delineating trends in fall-related nursing research to the present and directions of future research have significance not only for nursing studies but also for nursing practice.

This study explored the trends and patterns of fall-related nursing research by identifying the main topics of fall-related nursing research and changes over time. We used topic modeling to establish a nursing studies body of knowledge and present basic data for the development of fall-related nursing studies in the future. In particular, applying topic modeling in this study, which is the latest analysis technique in nursing [50], enabled a comprehensive understanding of current fall-related nursing research and core keywords as well as trends in research topics related to falls. These data may contribute to suggesting future directions for nursing research.

## 2. Materials and Methods

### 2.1. Study Design

This was a descriptive survey study that assessed the topics and trends of each main topic in fall-related nursing studies over time using topic modeling.

### 2.2. Subject Studies

Nursing studies related to patient falls were selected by searching international academic databases, including MEDLINE (PubMed), EMBASE (Excerpta Medica Database; Elsevier), Cochrane, CINAHL (Cumulative Index to Nursing and Allied Health Literature; EBSCO), Scopus, and Web of Science. Korean databases such as Research Information Sharing Service (RISS), National Digital Science Library (NDSL), and Korean Studies Information Service System (KISS) were searched for domestic studies. Relevant studies up to May 2020 were included. Studies that were published internationally (countries excluding South Korea) from 1974 and South Korean studies published from 1998 were included in this study. Those that were not journal articles (i.e., letters to editors, conference proceedings, and books) and whose abstracts were not available or written in languages other than English were excluded. In particular, non-fall-related international studies where the word “fall” was used in other contexts (i.e., fall asleep, fall in love, fall short, and sliding-scale insulin) were excluded.

### 2.3. Data Collection

English keywords were used in the searches, including “Accidental Falls”, “Fall”, “Falling”, “Slip*”, and “Slid*”, which were established by mixing natural language and medical subject heading (MeSH) terms; further, “And nurses*” was added to limit the search to fall-related nursing studies. The search content, document type, and language were limited to titles, articles, and English, respectively. Korean databases were searched using various combinations of “fall, nursing”, and “fall nursing”, and a literature search was performed with the advice of a nursing subject specialist librarian. The inclusion criteria were papers registered in domestic and international academic journals that reported nursing research related to falls. Exclusion criteria were papers with no abstracts and papers whose abstracts were not in Korean or English. Among the 6301 papers searched in international databases, 2450 papers were excluded and 3851 papers were reviewed based on their title and abstract. We then excluded 1274 papers not related to falls, 722 papers that were not articles, and 6 papers whose abstracts were not provided. Of the 570 papers searched in Korean databases, 309 papers were excluded and 261 papers were reviewed based on their title and abstract. We then excluded 28 not related to falls, 17 that were not journal articles, and 4 whose abstracts were not provided. The selection of articles was performed by a single researcher and a nursing professor who independently reviewed the titles and abstracts of the articles to exclude those not related to falls. A total of 2061 fall-related nursing studies, with 1849 international and 212 domestic studies, were included in the analysis of this study. The research procedure and flow of this study are shown in Figure 1.

### 2.4. Data Analysis

To prevent multiple selections of the same articles, bibliographic information—including authors’ names, publication year, title, and abstract—was exported to an Excel spreadsheet using EndNote X9 software. The abstract is the most likely section to contain keywords, other than the full text; therefore, the abstracts of the articles were analyzed according to a previous study [41]. Additionally, only nouns were extracted, according to a previous study, to facilitate the understanding of key concepts [51].

When texts are analyzed, it is necessary to refine the words as uppercase and lowercase letters, singular and plural, abbreviations, and special characters; in English, these differences can cause keywords to be classified as different words. Thus, dictionaries of designated keywords, synonyms, and excluded words were created to refine the words used for the analysis [52]. The dictionary of designated words allowed us to extract words in cases of multiple words having a single meaning [30,51]. Additionally, words that should be designated as proper or compound nouns such as “body mass index”, “Braden scale”, “pressure ulcer”, and “cardiovascular diseases” were organized in a dictionary. The dictionary of synonyms collated different words with the same or similar meanings and selected one representative word [30,53]; representative words of the MeSH terms were also organized in a dictionary by grouping words with similar meanings. For example, “Nurses”, which is one of the MeSH terms used in this study, was selected as the representative word for synonyms such as “nurse”, “nurses”, “Nurse”, “Nurses”, “registered nurse”, “registered nurses”, and “Registered Nurses”. The dictionary of excluded words removed one-letter words of which the meaning is difficult to understand and words that represent general concepts [30,54], while words related to research methodology (i.e., “background”, “method”, “result”, “discussion”, “conclusion”, “research”, and “study”) and statistics (i.e., “multiple regression analysis”, “Logistic regression analysis”, “correlation analysis”, and “multiple logistic regression”) were included. Moreover, special symbols and abbreviations were also included in the dictionary of excluded words.

Furthermore, frequency analysis was performed using NetMiner version 4.4 to extract words with a high frequency of appearance and influence from the fall-related nursing research. A total of 4142 words were extracted and refined. In this study, the frequency of a word is the number of its occurrences in the entire document. Core keywords were extracted by analyzing the top 20 words.

Topic modeling is an analysis method that estimates the probability of latent topics appearing in a document based on a document-word matrix [55]. For our study, a document(abstract)-word matrix was created to apply topic modeling. To perform topic modeling, the number of topics must first be determined to confirm the suitability of the research results and the possibility of interpretation through a Latent Dirichlet Allocation (LDA) algorithm using the NetMiner program version 4.4 [56]. Therefore, the number of topics was determined as the silhouette coefficient value through K-means clustering. K-means clustering minimizes the sum of squares of errors within a cluster. The number of topics and values of α and β—the LDA parameters—are used to select the number of topics with a silhouette coefficient value close to +1 [57]. Besides the top 10 words, those with the highest probability of appearance per topic were visualized as a topic–keyword map using a topic–word two-mode network. The topic group name was determined by referring to the top words for each topic.

To assess the changes in topics by period, the study timeline was divided into 10-year periods and the first, second, third, fourth, and fifth periods were 1974–1979, 1980–1989, 1990–1999, 2000–2009, and 2010–2020. IBM SPSS Statistics version 25.0 (SPSS, IBM Corporation) software was used to perform a linear regression analysis to identify topic types by assessing the patterns of increasing and decreasing trends by period. Each period was considered as an independent variable, while the weight of the entire document of each topic was considered as a dependent variable to establish a linear regression model. The independence of the assumption of the linear regression model was evaluated using a residual analysis and Durbin–Watson values [58]. Durbin–Watson values indicate the independence between the error terms after the regression analysis and range between 0 and 4; a value close to 2 indicates that the residuals are independent of each other and that the assumption of the linear regression model is satisfied [59].

The results of our linear regression analysis can be classified into four topic types, namely hot, warm, cool, and cold, according to the sign of the regression coefficient and the probability of significance. Topics with a positive coefficient sign and statistical significance, indicated by a significance probability of less than 0.05, were classified as “hot topics” with increasing research interest. Contrastingly, topics with a negative coefficient sign and statistical significance, indicated by a significance probability of less than 0.05, were classified as “cold topics” with research interest on the decline. Moreover, topics without statistical significance, indicated by a significance probability of greater than 0.05, were classified as “warm” and “cool” topics if the coefficient sign was positive or negative, respectively [60,61].

### 2.5. Ethical Considerations

This study was exempted from approval by the institutional review board of the first author’s affiliate university (IRB No. 1041078-202005-HRSB-132-01).

## 3. Results

### 3.1. Core Keywords

A total of 4142 words were extracted and refined from the abstracts of 2061 papers. The 20 most frequently occurring keywords in fall-related nursing research are shown in Table 1. The simple frequency of words was as follows: “medication” (1166 times), “fall prevention” (954), “fall injury” (933), “older adult” (802), “educational program” (765), “fracture” (744), “quality” (665), “restraint” (647), “patient safety” (611), and “health” (609).

### 3.2. Topic Modeling

The number of topics for fall-related nursing studies was determined based on the extracted silhouette coefficient, which was calculated using the number of topics and LDA α and β. An upper silhouette coefficient value of 0.939 was calculated with four topics and LDA parameters of α = 0.2 and β = 0.02; thus, four topics were modeled. The keywords of fall-related nursing studies by topic are provided in Table 2.

Topic 1 accounted for 20.28% of all articles, and the top 10 keywords—in order of decreasing probability—were “balance”, “older adult”, “exercise program”, “test”, “fall prevention”, “fear of falling”, “educational program”, “scale”, “physical activity”, and “activity of daily living”, with respective probabilities of 0.032, 0.028, 0.021, 0.019, 0.019, 0.019, 0.018, 0.016, 0.016, and 0.015. Topic 1 was named “fall prevention program and scale” and included papers that improved balance by promoting physical activity and activities of daily living in older adults, developed exercise or education programs to decrease the fear of falling, performed tests, and verified the validity of scales used to assess risk factors.

Topic 2 accounted for 43.72% of all the articles, and the top 10 keywords—in order of decreasing probability—were “fall prevention”, “quality”, “patient safety”, “educational program”, “nursing process”, “fall knowledge”, “implementation”, “health”, “strategy”, and “improvement”, with respective probabilities of 0.023, 0.019, 0.019, 0.017, 0.013, 0.011, 0.010, 0.010, 0.009, and 0.008. Topic 2 was named “nursing strategy for fall prevention” and included studies that provided strategies to increase the quality of nursing and the implementation of patient safety measures to improve subjects’ health problems by developing educational programs and fall-related nursing guidelines to develop the knowledge of falls when nursing processes to prevent falls were applied.

Topic 3 accounted for 17.13% of the studies, and the top 10 keywords—in order of decreasing probability—were “fracture”, “fall injury”, “hospitalization”, “stroke”, “cost”, “treatment”, “osteoporosis”, “death”, “diagnosis”, and “discharge”, with respective probabilities of 0.050, 0.040, 0.028, 0.015, 0.014, 0.014, 0.013, 0.011, 0.010, and 0.010. Topic 3 was referred to as “hospitalization by fall injury” and included studies related to treatment duration, delayed discharge, increased cost, and death after diagnosis of fracture due to fall injury and hospitalization in patients with risk factors for falls, such as osteoporosis and stroke.

Topic 4 accounted for 18.87% of the data, and the top 10 keywords—in order of decreasing probability—were “medication”, “restraint”, “dementia”, “long-term care facility fall”, “adverse effects”, “association”, “fall injury”, “older adult”, “problem”, and “symptom”, with respective probabilities of 0.069, 0.040, 0.031, 0.019, 0.018, 0.014, 0.012, 0.012, 0.011, and 0.009. Topic 4 was named “long-term care facility fall” and included studies on the symptoms of dementia in older adults hospitalized in long-term care facilities, long-term care facility falls associated with the adverse effects of medication treatment, and interventions such as medication treatment and restraint to improve health problems and prevent fall injuries.

The results of visualizing the network of ten keywords in the upper probability distribution of the four main topics using the topic–keyword map are shown in Figure 2. “Educational program” and “fall prevention” were related to Topics 1 and 2 and were included in the keywords, with top 10 probability distributions in both topics. “Older adult” was related to Topics 1 and 4 and was one of the keywords with top 10 probability distributions in both topics. Moreover, “fall injury” was related to Topics 3 and 4 and was one of the keywords with top 10 probability distributions in both topics. A word cloud according to the frequency of words for each topic is shown in Figure 3. A word cloud, which is an image created using the NetMiner program version 4.4 to convey words that are used most frequently in a work or body of works, for each topic is shown in Figure 3.

### 3.3. Topic Trend

A total of 2061 articles were divided into 10-year intervals, and changes in the topics over time were assessed. A total of 6, 24, 204, 654, and 1173 articles were included in Periods 1, 2, 3, 4, and 5, respectively.

The changes in the proportion of fall-related nursing studies by topic over time are shown in Figure 4. Topic 1 (fall prevention program and scale) accounted for 16.67% of all articles in Period 2 with four articles. In Period 3, the distribution was 7.35% with 16 articles; moreover, increasing trends were shown with 18.81% (123 articles) and 23.53% (276 articles) in Periods 4 and 5, respectively. Topic 2 (nursing strategy for fall prevention) accounted for 66.67% of all articles in Period 1 with four articles. In Period 2, its proportion was 62.50% with 15 articles, with 50.98% (104 articles) in Period 3; this topic showed a decreasing trend with 45.26% (296 articles) and 41.09% (482 articles) in Periods 4 and 5, respectively. Topic 3 (hospitalization by fall injury) accounted for 16.67% of all articles in Period 1 with one article. In Period 2, the proportion decreased to 12.50% with three articles, and in Period 3, it was 20.59% with 42 articles. In Periods 4 and 5, a decreasing trend was observed with 16.97% (111 articles) and 16.71% (196 articles), respectively. Topic 4 (long-term care facility fall) accounted for 16.67% of all articles in Period 1 with one article. In Period 2, a distribution of 8.33% (2 articles) was observed, which increased to 21.08% with 43 articles in Period 3. However, Periods 4 and 5 showed a decreasing trend, with 18.96% (124 articles) and 18.67% (219 articles) in Periods 4 and 5, respectively.

The trends of fall-related nursing studies by topic are shown in Table 3. The four main fall-related nursing study topics were classified according to their regression coefficients and the significance probabilities of the linear regression analysis. The Durbin–Watson value of residual analysis was calculated to assess the independence of the linear regression model; this value was close to 2 in all four topics, indicating that the assumption of independence of the linear regression model was satisfied [51]. Topic 1 (fall prevention program and scale) was classified as a “hot topic”, as it was statistically significant and had a positive regression coefficient (B = 0.511, t = 3.422, *p* = 0.001). Topic 2 (nursing strategy for fall prevention) was classified as a “cold topic”, as it was statistically significant and had a negative regression coefficient (B = −0.748, t = −3.244, *p* = 0.002). However, Topic 3 (hospitalization by fall injury) was not statistically significant and had a negative regression coefficient. Therefore, it was classified as a “cool topic” (B = −0.099, t = −0.491, *p* = 0.626), while Topic 4 was classified as a “hot topic” for having statistical significance and a positive regression coefficient (B = 0.337, t = 2.746, *p* = 0.009).

## 4. Discussion

This study analyzed the abstracts of fall-related nursing studies and assessed the core keywords, main topics, and trends, as well as the pattern of each topic over time using topic modeling. As a result of assessing the frequency of keywords in fall-related nursing research, meaningful words among the top 20 were “fracture” and “fall risk assessment tool”.

Fracture is one of the common physical injuries caused by falls [62] and mainly occurs in elderly people with weakened muscle and body conditions due to decreased bone mass associated with aging [63]. Falls not only cause socioeconomic burdens, such as physical damage, emotional impact, prolonging of hospital stay, and medical expenses [63,64] but are also decisive factors in osteoporotic fractures. As a strategy for preventing osteoporotic fractures [65], falls must be addressed as a core goal of nursing research. Fall risk assessment tools have been developed and used in a variety of clinical settings, and accurate prediction of high-risk patients is fundamental in fall prevention activities [66]. Risk assessment tools must identify fall risk patients and thereby enable systematic management to prevent falls; this is a necessary focus for nursing research.

Topic 1 (fall prevention program and scale) was classified as a hot topic with gradually increasing interest over time; the studies on this topic focus on exercise and education programs as interventions for fall prevention. In 2011, the American Geriatrics Society emphasized the assessment of fall risk factors, multi-faceted interventions, and environment adjustments to prevent falls [67]. It was observed that multi-faceted programs consisting of exercise therapy and education on exercise, nutrition, and environmental adjustment were effective in preventing falls [68]. A systematic literature review study also suggested the necessity of studies on complex programs that consider physical factors such as balance improvement and psychological factors including depression and cognitive levels [69]. Studies on multi-faceted interventions that comprehensively consider physical factors such as improving balance and muscle strength, psychological factors such as cognitive decline and depression, and environmental adjustment would be necessary [70].

The fall guidelines published by the Registered Nurses’ Association of Ontario (RNAO) highlighted using tools that consider both the subjects and the environment [71]. A systematic review of international studies also suggested that an assessment tool that considers the age and characteristics of the subjects needs to be developed [72]. In a study published in South Korea, it was suggested that the Morse fall assessment tool is limited to the assessment of subjects’ physical factors and that a fall risk assessment tool that considers the cognitive levels and individual characteristics of the subjects and is suitable for environmental characteristics needs to be developed [73]. Studies have shown that fall risk assessment tools do not reflect the characteristics of the group with a high risk of falling and that the scores—which serve as the criteria for the high-risk group—did not predict the actual risk of falling [74]. Moreover, as studies have identified various risk factors for falls—such as osteoporosis, muscle loss, balance, cognitive decline, and the environment [75,76]—it is thought that these studies are limited in their applicability to clinical settings. Therefore, further studies should be performed to develop fall risk assessment tools suitable for the current healthcare system in South Korea and consider the individual characteristics of subjects. Further, translational studies to apply findings of fall risk factors in clinical settings must be conducted.

Topic 2 (nursing strategy for fall prevention) was a cold topic that showed a gradually decreasing trend over time; however, Topic 2 still accounts for a high distribution of all topics. Various nursing strategies have been developed [6,77,78,79], and, as falls are caused by a combination of factors, nursing strategies that fit the trend and the current environment need to be developed. Fall prevention nursing guidelines in other countries emphasize the importance of policies and support within hospitals, as well as the association between the systems of different departments. In contrast, fall prevention nursing guidelines in South Korea focus on fall nursing practices for individual patients [6]. In a study published in South Korea that assessed nurses’ experiences of fall accidents [73], improvements in environmental factors—such as slippery floors, insufficient lighting, bed height, bed railing, and floor shock absorbers—were suggested to prevent falls and reduce fall-related injuries. Moreover, there is a need for systematic support to protect patients from falls and the recruitment of nursing staff to increase the quality and hours of nursing. The fall practice guidelines of the UK Emergency Care Research Institute also emphasize the importance of establishing fall management teams and policies in medical institutions [80]. Therefore, research on the development and establishment of systems and policies related to fall prevention and management in medical institutions is necessary for South Korea.

Topic 3 (hospitalization by fall injury) was a cool topic with a gradually decreasing trend over time. It could be that the number of studies on the hospitalization outcomes of fall-related injuries is decreasing because studies have been focusing on fall prevention after the Centers for Medicare and Medicaid Services suggested that falls are preventable [17]. Fall-related injuries require additional diagnostic tests, surgeries, and treatment and have a high recurrence rate of fractures, leading to long-term hospitalization and high medical costs. Falls, therefore, have a negative effect not only on patients and guardians but also socially and economically on the community [81]. Existing studies on rehabilitation treatment for fractures caused by falls [82], community preventive interventions and connections with long-term care [83], and multi-disciplinary approaches have been performed in other countries. In contrast, the number of studies on systems, such as the integrated association between related community organizations [84], is decreasing in South Korea [85]. The fall practice guideline published by the Agency for Healthcare Research and Quality emphasized cooperation between multi-disciplinary teams for the prevention and management of falls [86]; moreover, Oh reported that rehabilitation treatment after the fracture surgery should aim to promote the early recovery of gait ability and sense of balance and that multi-disciplinary cooperation is necessary to prevent falls [87]. Therefore, studies on the rehabilitation treatment for fall-related injuries in hospitalized patients, the connection to local communities, and a multi-disciplinary approach are necessary for South Korea. Moreover, studies on falls have been biased toward economic perspectives such as the length of stay, mortality, and medical costs of hospitalized patients in medical institutions [88]. Thus, falls reflect medical accidents and the strict standards for the quality of patient safety in medical institutions [89,90], and studies on the rights, dignity, and safety of individuals who experience falls should be developed in the future.

Topic 4 (long-term care facility fall) was classified as a hot topic with a gradually increasing trend. As the older population has multiplied, the number of older adults with dementia is also increasing [91]. Although studies on the relationship between falls and older adults with dementia and cognitive impairment have been performed in other countries [92,93], there is a lack of fall-related studies involving older individuals with dementia in South Korea. Previous studies on falls among older adults with dementia and cognitive impairment have been performed; however, only a limited number of studies have been reported, and most studies were conducted on older adults in local communities [63]. A literature review on interventions for fall prevention in older adults with dementia reported that the individual risk factors of falls need to be assessed to prevent falls [94], with an interest in the therapeutic and adverse effects of medications that older adults with dementia are consuming, and that assistive devices and equipment appropriate for each individual must be used. The study further suggested that exercise interventions, cognitive behavioral therapy, and therapeutic activities appropriate to the physical characteristics of subjects who are vulnerable to falls should be provided. The fall practice guidelines published by Veteran Affairs Healthcare in the United States also highlighted the assessment of individual fall risk factors and the management of vulnerable patients to prevent falls in long-term care facilities [95]. As the proportion of adults older than 65 and the number of older adults with dementia is expected to exceed 20% in 2025 and increase to 1.27 million by 2030, studies on the assessment of individual risk factors for falls and the management of high-risk patients with dementia would need to be conducted in the future [96,97,98].

### 4.1. Implications for the Regulators and Future Research Agenda

When comprehensively considering trends in nursing research related to falls, falls as an indicator of the quality of patient safety in nursing has been developed and evaluated in various ways, such as for assessment tools, education and exercise programs, and nursing guidelines to prevent falls. However, there are very few translational studies to enable the prior studies to affect fall-related outcomes in clinical nursing, such as fall occurrence and fall injuries. Translational research applies basic and clinical research results as practice guidelines and protocols, compares and analyzes the effectiveness of existing and new practice guidelines, and provides education and training according to the situation in the clinical field. All relevant studies are referenced for the realization and maintenance of new working guidelines [99,100]. Therefore, in the future, it is necessary to conduct translational studies of falls in a nursing context, which would reflect fall risk factors, fall risk assessment tools, and fall prevention programs in the actual working environment.

### 4.2. Limitations

This study has certain limitations that need to be mentioned. First, to establish a search strategy that can be objectively verified at the data collection stage, we strategically searched for fall-related nursing studies with the advice of a nursing subject librarian, and only those studies in which agreements were reached between one professor and a researcher were collected to exclude studies that have used the word “fall” for different meanings. Therefore, there is a limitation that nursing studies related to falls may have been excluded from the analysis.

Second, MeSH terms were used to extract and refine words, which are the basic units of text network analysis, and words with limited MeSH terms were refined to words observed in previous studies. However, the words were expressed in various manners by different authors, and, therefore, bias in the extraction of keywords from the abstract cannot be excluded.

## 5. Conclusions

This study is meaningful since it presents trends in nursing research related to falls through topic modeling that may be used as basic data for future research topic development. Further research is needed to protect the safety, dignity, and rights of people at high risk and to establish policies and system development regarding fall prevention and management in medical institutions in Korea, as research in this area is insufficient compared to other countries. Finally, to prevent falls by applying existing fall-related nursing studies to actual clinical practice, it is necessary to conduct a translational study of fall-related nursing reflected in actual practice through education and training suitable for the nursing field.

## Figures and Tables

**Figure 1 ijerph-18-03963-f001:**
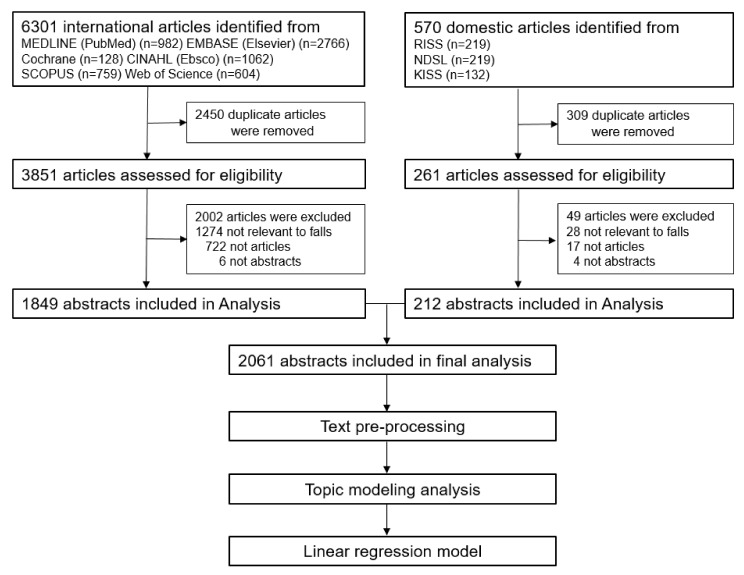
Research flow.

**Figure 2 ijerph-18-03963-f002:**
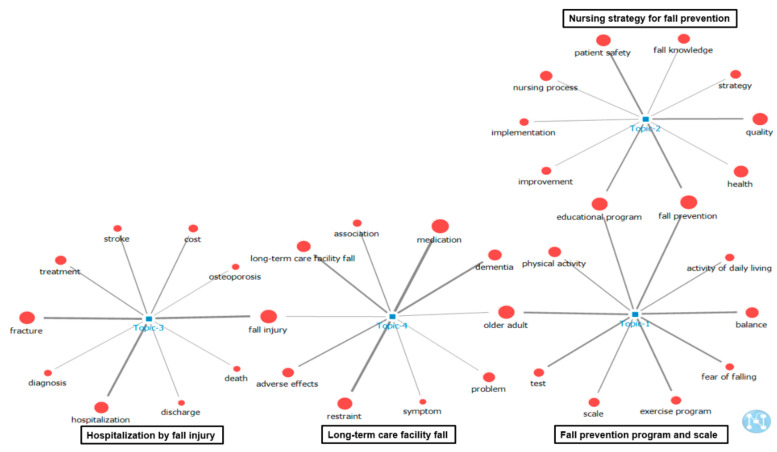
Two-mode analysis of topic–keyword.

**Figure 3 ijerph-18-03963-f003:**
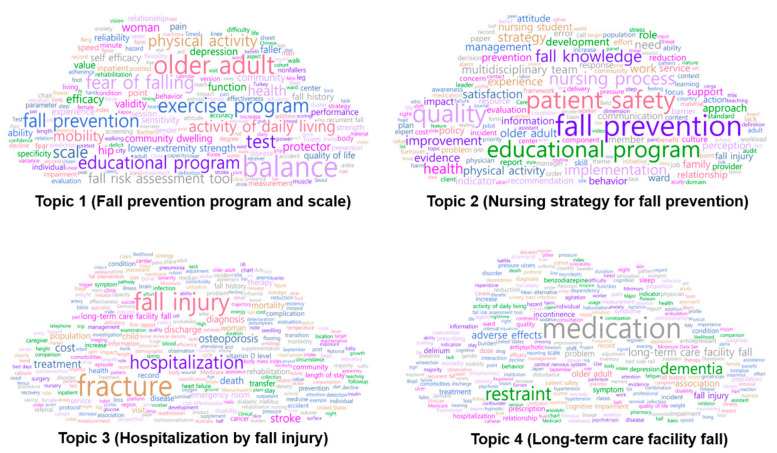
Word clouds by topic.

**Figure 4 ijerph-18-03963-f004:**
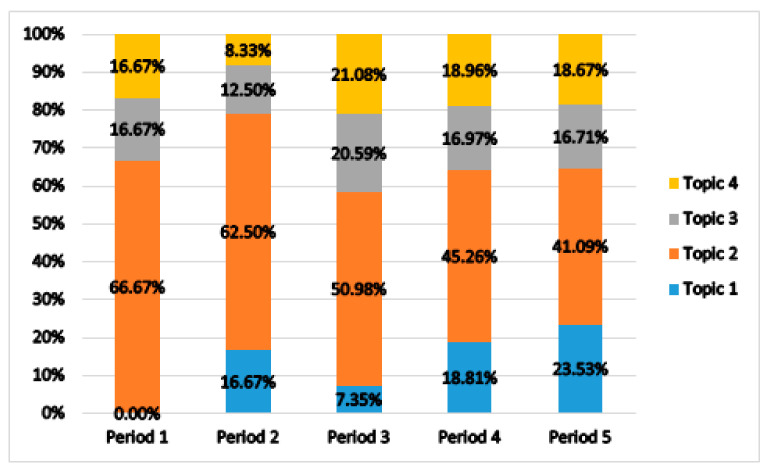
Topic trend by period.

**Table 1 ijerph-18-03963-t001:** Core keywords by frequency.

Rank	Keyword	Frequency	Rank	Keyword	Frequency
1	medication	1166	11	hospitalization	549
2	fall prevention	954	12	long-term care facility fall	525
3	fall injury	933	13	dementia	506
4	older adult	802	14	balance	471
5	educational program	765	15	physical activity	441
6	fracture	744	16	fall risk assessment tool	386
7	quality	665	17	nursing process	373
8	restraint	647	18	problem	364
9	patient safety	611	19	fall knowledge	363
10	health	609	20	adverse effects	359

**Table 2 ijerph-18-03963-t002:** Topic group and high-ranking keywords.

Category	Topic 1		Topic 2		Topic 3		Topic 4	
Topic group	fall prevention program and scale		nursing strategy for fall prevention		hospitalization by fall injury		long-term care facility fall	
Proportion	20.28%		43.72%		17.13%		18.87%	
High-ranking keywords	balance	0.032	fall prevention	0.023	fracture	0.050	medication	0.069
	older adult	0.028	quality	0.019	fall injury	0.040	restraint	0.040
	exercise program	0.021	patient safety	0.019	hospitalization	0.028	dementia	0.031
	test	0.019	educational program	0.017	stroke	0.015	long-term care facility fall	0.019
	fall prevention	0.019	nursing process	0.013	cost	0.014	adverse effects	0.018
	fear of falling	0.019	fall knowledge	0.011	treatment	0.014	association	0.014
	educational program	0.018	implementation	0.010	osteoporosis	0.013	fall injury	0.012
	scale	0.016	health	0.010	death	0.011	older adult	0.012
	physical activity	0.016	strategy	0.009	diagnosis	0.010	problem	0.011
	activity of daily living	0.015	improvement	0.008	discharge	0.010	symptom	0.009

**Table 3 ijerph-18-03963-t003:** Topic type by regression coefficient and significance probability.

Category	B	β	t	*p*-Value	Durbin–Watson	Topic Type
Topic 1 (Fall prevention program and scale)	0.511	0.481	3.422	0.001	1.443	Hot
Topic 2 (Nursing strategy for fall prevention)	−0.748	−0.461	−3.244	0.002	2.491	Cold
Topic 3 (Hospitalization by fall injury)	−0.099	−0.078	−0.491	0.626	2.518	Cool
Topic 4 (Long-term care facility fall)	0.337	0.403	2.746	0.009	1.569	Hot

## Data Availability

No new data were created or analyzed in this study. Data sharing is not applicable to this article.

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
