# Peer review of "Trends of Nursing Research on Accidental Falls: A Topic Modeling Analysis"

_ijerph, 2021, doi:10.3390/ijerph18083963_

Round 1

Reviewer 1 Report

Thank you very much for the opportunity to review this manuscript. The manuscript identifies the popular topic of nursing research on accidental falls via topic modelling. The manuscript is well structure. However, I have the following comments to improve this paper:

  • The authors add more specific implications of the study in the abstract.
  • The study's objective must be moved up in the first paragraph.
  • The authors highlight the study's incremental contributions compared to prior relevant studies in the last paragraph of the introduction.
  • Authors cite recent articles in the introduction published in 2021.
  • The authors add specific inclusion criteria in section 2.3.
  • Authors insert a table to describe the total number of words, countries, number of keywords, and any other important information about their sample.
  • Authors add a sub-section 4.1 in discussion with “Implications for the regulators and future research agenda” as these are main outcome of their research.
  • Authors present the figures with high quality for better demonstration.
  • Finally, the authors check the minor issues and typos.

Author Response

Reviewer: 1

Thank you very much for the opportunity to review this manuscript. The manuscript identifies the popular topic of nursing research on accidental falls via topic modelling. The manuscript is well structure. However, I have the following comments to improve this paper:

  1. The authors add more specific implications of the study in the abstract.

Response: As per your comments, we have revised the Abstract accordingly.

Abstract, Lines 11–13: We extracted only nouns from each abstract, and four topics were identified through topic modeling, which were divided into aspects of fall prevention and its consequences.

Abstract, Lines 21–23 This study confirms the trends in domestic and international fall-related research, suggesting the need for studies to address insufficient fall-related policies and systems and translational research to be applied in clinical trials.

  1. The study's objective must be moved up in the first paragraph.

Response: As per your comment, we moved the text accordingly.

Page 1, Lines 37–40: Moreover, in attempts to prevent falls due to these socioeconomic issues, fall-related nursing research has been extensively carried out to date. This study details the current status of fall-related nursing research and its trends over time.

  1. The authors highlight the study's incremental contributions compared to prior relevant studies in the last paragraph of the introduction.

Response: We added applicable text, accordingly.

Page 2, Lines 88–92: In particular, applying topic modeling in this study, which is the latest analysis technique in nursing [50], enabled comprehensive understanding of current fall-related nursing research and core keywords, besides trends in research topics related to falls. These data may contribute to suggesting future directions for nursing research.

  1. Authors cite recent articles in the introduction published in 2021.

Response: As per your comment, we have cited articles published in 2021, such as Chen and Shin, Panneman et al., and Vlaeyen et al.

  1. The authors add specific inclusion criteria in section 2.3.

Response: We added these criteria accordingly.

Page 3, Lines 119–122: The inclusion criteria were papers registered in domestic and international academic journals that reported nursing research related to falls. Exclusion criteria were papers with no abstracts and papers whose abstracts were not in Korean or English.

  1. Authors insert a table to describe the total number of words, countries, number of keywords, and any other important information about their sample.

Response: We now present the word refining process before topic modeling was performed. We also present the total number of words, and key words according to frequency analysis are additionally presented in the results (Table 1).

Page 5, Lines 163–167: Furthermore, frequency analysis was performed using the NetMiner version 4.4 to extract words with a high frequency of appearance and influence from the fall-related nursing research. A total of 4,142 words were extracted and refined. In this study, the frequency of a word is the number of its occurrences in the entire document. Core keywords were extracted by analyzing the top 20 words.

Page 6, Lines 208–215

3.1. Core Keywords

A total of 4,142 words were extracted and refined from the abstracts of 2,061 papers. The 20 most frequently occurring keywords in fall-related nursing research are shown in Table 1. The simple frequency of words was as follows: “medication” (1166 times), “fall prevention” (954), “fall injury” (933), “older adult” (802), “educational program” (765), “fracture” (744), “quality” (665), “restraint” (647), “patient safety” (611), and “health” (609).

Table 1. Core keywords by frequency

Rank

Keyword

Frequency

Rank

Keyword

Frequency

1

medication

1166

11

hospitalization

549

2

fall prevention

954

12

long-term care facility fall

525

3

fall injury

933

13

dementia

506

4

older adult

802

14

balance

471

5

educational program

765

15

physical activity

441

6

fracture

744

16

fall risk assessment tool

386

7

quality

665

17

nursing process

373

8

restraint

647

18

problem

364

9

patient safety

611

19

fall knowledge

363

10

health

609

20

adverse effects

359

  1. Authors add a sub-section 4.1 in discussion with “Implications for the regulators and future research agenda” as these are main outcome of their research.

Response: As per your comments, we have rechecked and revised it accordingly.

Page 13, Lines 422–435: 4.1. Implications for the regulators and future research agenda

When comprehensively considering trends in nursing research related to falls, falls as an indicator of the quality of patient safety in nursing have been developed and evaluated in various ways, such as for assessment tools, education and exercise programs, and nursing guidelines to prevent falls. However, there are very few translational studies to enable the prior studies to affect fall-related outcomes in clinical nursing, such as fall occurrence and fall injuries. Translational research applies basic and clinical research results as practice guidelines and protocols, compares and analyzes the effectiveness of existing and new practice guidelines, and provides education and training according to the situation in the clinical field. All relevant studies are referenced for the realization and maintenance of new working guidelines [99,100]. Therefore, in the future, it is necessary to conduct translational studies of falls in a nursing context, which would reflect fall risk factors, fall risk assessment tools, and fall prevention programs in the actual working environment.

  1. Authors present the figures with high quality for better demonstration.

Response: As per your comments, we have improved the quality of the figures.

Page 4, 9, 10: Figure 1, 2, 3, 4

  1. Finally, the authors check the minor issues and typos.

Response: We have rechecked the text and corrected any minor typos as necessary.

Reviewer 2 Report

Dear authors,

First of all, congratulations on your work.

The topic in question has been continuously addressed in the literature. This has made your approach very interesting, as we can see the trends in how fall prevention topics have changed through time. Overall I have no major concerns. Recommendations would be:

Abstract: Try providing an abstract that targets a wider audience, as it is too dense and with many results. A more synthesized and direct to the point abstract would help the reader understand the importance of your work.

Introduction: Lines 65-75 would benefit from better integration of the main topic (falls) to explain the methodology. How it is presented deviates from the main topic. It would be more suitable in the material and methods rather than in the introduction.

2.2 - Subject studies: Which database was searched on PubMed? MEDLINE? This is important as Pubmed is a platform/search engine. If it is MEDLINE, I would recommend displaying it as: "MEDLINE (PubMed)". The same recommendation applies to CINAHL (e.g., "CINAHL (Ebsco)").

Conclusion: The conclusion could have a "wow" factor, specifically in your results' importance. Not that the findings are adequate (quite the opposite), but they are presented in a telegraphic form.

As stated, these are only minor recommendations that could introduce more interest in an already interesting article.

Author Response

Reviewer: 2

First of all, congratulations on your work.

The topic in question has been continuously addressed in the literature. This has made your approach very interesting, as we can see the trends in how fall prevention topics have changed through time. Overall I have no major concerns. Recommendations would be:

  1. Abstract: Try providing an abstract that targets a wider audience, as it is too dense and with many results. A more synthesized and direct to the point abstract would help the reader understand the importance of your work.

Response: As per your comment, we have modified the Abstract as follows.

Abstract, Lines 11–13: We extracted only nouns from each abstract, and four topics were identified through topic modeling, which were divided into aspects of fall prevention and its consequences.

Abstract, Lines 21–23 This study confirms the trends in domestic and international fall-related research, suggesting the need for studies to address insufficient fall-related policies and systems and translational research to be applied in clinical trials.

  1. Introduction: Lines 65-75 would benefit from better integration of the main topic (falls) to explain the methodology. How it is presented deviates from the main topic. It would be more suitable in the material and methods rather than in the introduction.

Response: As per your comment, we have altered the text accordingly.

Page 2, Lines 77–84:

A fall is a sudden complex event that occurs by combining intrinsic and extrinsic factors that affects many aspects of individuals, society, and local communities [1,34]. It will be effective to analyze the flow and trends of big data of fall-related nursing research over time using topic modeling. Considering that studies related to social interest in falls will continue to increase in the future [47-49], delineating trends in fall-related nursing research to the present and directions of future research have significance not only for nursing studies but also for nursing practice.

  1. Subject studies: Which database was searched on PubMed? MEDLINE? This is important as Pubmed is a platform/search engine. If it is MEDLINE, I would recommend displaying it as: "MEDLINE (PubMed)". The same recommendation applies to CINAHL (e.g., "CINAHL (Ebsco)").

Response: As per your comments, we have explained the database search more precisely.

Page 3, Lines 99–102: Nursing studies related to patient falls were selected by searching international academic databases, including MEDLINE (PubMed), EMBASE (Excerpta Medica Database), Elsevier, Cochrane, CINAHL (Cumulative Index to Nursing and Allied Health Literature; Ebsco), Scopus, and Web of Science.

  1. Conclusion: The conclusion could have a "wow" factor, specifically in your results' importance. Not that the findings are adequate (quite the opposite), but they are presented in a telegraphic form.

Response: As per your comments, we have enhanced the conclusion accordingly.

Page 14, Lines 450–457: This study is meaningful since it presents trends in nursing research related to falls through topic modeling that may be used as basic data for future research topic development. Further research is needed to protect the safety, dignity, and rights of persons at high risk and to establish policies and system development regarding fall prevention and management in medical institutions in Korea, which is insufficient compared to other countries. Finally, to prevent falls by applying existing fall-related nursing studies to actual clinical practice, it is necessary to conduct a translational study of fall-related nursing reflected in actual practice through education and training suitable to the nursing field.

  1. As stated, these are only minor recommendations that could introduce more interest in an already interesting article.

Response: Thank you for your positive comments.

Round 2

Reviewer 1 Report

Accept in present form.